# Influence of glucose on swarming and quorum sensing of *Dickeya solani*

**Roberta Gatta[1], Andrzej Wiese[1], Adam Iwanicki[2], Michał Obuchowski** [ID][2]*

**1** Division of Medical Biotechnology, Intercollegiate Faculty of Biotechnology UG-MUG, University of Gdańsk, Gdańsk, Poland, **2** Division of Medical Biotechnology, Intercollegiate Faculty of Biotechnology UG-MUG, Medical University of Gdańsk, Gdańsk, Poland

* michal.obuchowski@ug.edu.pl

**Data Availability Statement:** All relevant data are within the paper.

**Funding:** This work was supported by the National Science Center OPUS project no. 2018/29/B/NZ9/02339. The funders had no role in study design,

## Abstract

*Dickeya solani* is a pathogen most frequently responsible for infecting potato plants in Europe. As in the case of most plant pathogens, its ability to colonize and invade the host depends on chemotaxis and motility. The coordinated movement of *Dickeya* over solid surfaces is governed by a quorum sensing mechanism. In *D. solani* motility is regulated by ExpI-ExpR proteins, homologous to *luxI-luxR* system from *Vibrio fisheri*, in which *N*-acyl-homoserine lactones (AHLs) serve as signaling molecules. Moreover, in many Gram-negative bacteria motility is coupled with central metabolism via carbon catabolite repression. This enables them to reach more nutrient-efficient niches. The aim of this study was to analyze the swarming motility of *D. solani* depending on the volume of the medium in the cultivation plate and glucose content. We show that the ability of this bacterium to move is strictly dependent on both these factors. Moreover, we analyze the production of AHLs and show that the quorum sensing mechanism in *D. solani* is also influenced by the availability of glucose in the medium and that the distribution of these signaling molecules are different depending on the volume of the medium in the plate.

## Introduction

*Dickeya* genus consists of pathogens targeting economically important plants. According to the current classification, the genus includes the following species: *D. aquatica*, *D. chryzanthemi*, *D. dadanti*, *D. dianthicola*, *D. fangzhongdai*, *D. paradisiaca*, *D. solani*, and *D. zeae* [1–5]. Among them, *Dickeya solani* species are bacteria most commonly infecting potato plants in Europe causing black-leg and soft rot diseases [6].

In its natural environment, *Dickeya* must deal with limited nutrients availability and stressful conditions encountered in water and soil. Because of that, the pathogenesis of these bacteria must be strictly controlled to avoid waste of energy and unnecessary production of virulence factors. So far 10 major regulators of *Dickeya* virulence have been described: KdgR, PecS, PecT, CRP, H-NS, Fis, Fur, GacA, SlyA and MfbR [7, 8]. There are two phases of *Dickeya* pathogenic cycle. In the first bacteria remain latent while penetrating the host through wounds or natural openings [9]. Upon colonization, bacteria multiply slowly while constantly monitoring the availability of plant soluble sugars which can be used in cellular metabolism. Along with

data collection and analysis, decision to publish, or preparation of the manuscript.

**Competing interests:** The authors have declared that no competing interests exist.

the increasing availability of such sugars bacteria switch to the second phase of the pathogenic cycle in which they multiply massively and undergo profound metabolic changes. They start producing large amounts of plant cell wall-degrading enzymes (CWDEs), mainly pectate lyases, which provide bacteria with required nutrients and lead to the destruction of plant tissues [10].

An essential role in the colonization of plants by these pathogens is played by chemotaxis and motility. These two phenomena enable bacteria to sense compounds of plant origin and move towards their source which is most probably a wound or natural opening in the plant tissue. Bacterial movement over solid surface occurs by the mechanism of swarming [11] which, as a coordinated social behavior of bacteria, must be subjected to regulation. In *D. solani* and many other bacteria, this phenomenon is regulated by a quorum sensing mechanism which serves as a cell-to-cell communication system [12].

There are two quorum sensing systems found in members of the *Dickeya* genus. The first one is the classic system mediated by *N*-acyl-homoserine lactones (AHLs) and is homologous to the best-studied *luxI-luxR* system from *Vibrio fisheri* [13]. In *Dickeya* this system consists of ExpI-ExpR proteins [14] and is known to regulate protease production and motility of these bacteria [15]. Exp system relies mainly on *N*-(3-oxohexanolyl)-homoserine lactone with only minimal contribution of *N*-(hexanoyl)-homoserine lactone [16]. The second system is called Virulence Factor Modulating (VFM) and is encoded in 25 kb cluster of genes whose products are involved in the pathogenesis and production of CWDEs [17]. VFM was shown in *Dickeya* to be unrelated to motility [15].

Swarming was shown to be dependent on the presence of catabolite repression protein (Crc) in *Pseudomonas syringae*, another Gram-negative plant pathogen [18]. The canonical function of carbon catabolite repression is directing the choice of nutrients by favoring the utilization of the most efficiently metabolizable sugars [19]. Such coupling with the central metabolism suggests that the motility of bacteria is associated with the metabolic state of the cell and availability of nutrients in the environment helping bacteria in reaching more nutrient-efficient niches. In *Dickeya* no such direct link has been shown, nonetheless, the expression of virulence genes is known to be activated when efficient carbon sources are exhausted. This regulation occurs mostly by the carbon repression protein (CRP) [20].

In this study, we investigated how the volume and composition of cultivation media influenced the swarming of *D. solani*. We also analyzed how the concentration of glucose in the medium impacts the production of quorum sensing molecules by these bacteria.

## Results

Triggering swarming motility of bacteria in the laboratory requires precisely controlled conditions such as medium composition, percentage of agar, time of plate drying, temperature and humidity at incubation. Most publications concerning the swarming of *Dickeya* species provide general information regarding media composition and percentage of agar which usually is 0.5%. We have used this concentration of agar and performed swarming assays with changing volumes of the medium used in Petri plates. To our surprise, *Dickeya* was unable to swarm on plates containing more than 10 ml of solid medium (Fig 1).

The most efficient swarming was observed in the case of plates containing 5 ml of medium, nevertheless, due to difficulties with the preparation of uniform plates (very fast solidification of agar), we decided to use plates containing 7.5 ml of medium in further experiments.

The process of surface colonization and biofilm formation in Gram-negative plant pathogens is coordinated via a quorum sensing system [12]. We wanted to check whether observed differences in swarming of *Dickeya* correlated with the presence of *N*-acyl homoserine lactones

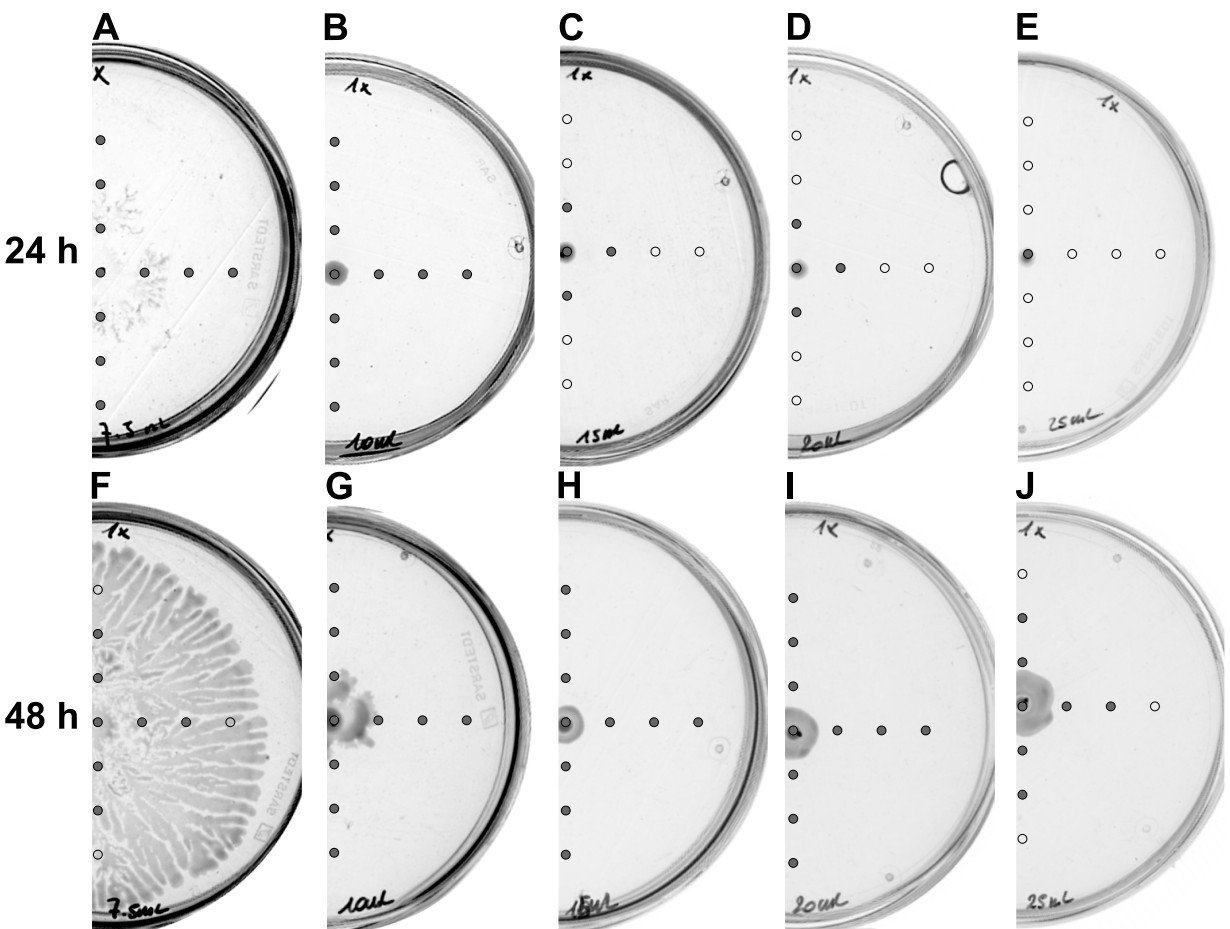

**Fig 1.** Swarming and AHLs production by *Dickeya solani* IFB102 on plates containing following volumes of B-medium: (A, F) 7.5 ml, (B, G) 10 ml, (C, H) 15 ml, (D, I) 20 ml, (E, J) 25 ml. Swarming assays were performed for 24 and 48 h, as indicated. Grey circles represent spots at which AHLs were detected. Open circles represent spots at which no AHLs were detected.

(AHLs) in the medium. AHLs are common diffusible signaling molecules utilized by Gram-negative bacteria for this purpose [21]. We performed swarming assays on plates containing different volumes of the medium. Medium in the plates was then sampled at several spots and tested for the presence of AHL molecules using a biosensor strain of *E. coli* PSB401 [22]. After 24 hours of incubation, we were able to detect AHLs at all tested spots of 7.5 and 10 ml plates (Fig 1A and 1B), suggesting that these molecules were present in the entire plates. In the case of plates containing 15 and 20 ml of the medium we were able to detect these quorum sensing molecules at the inoculation spot and within 1 cm radius from it (Fig 1C and 1D). In the plates containing 25 ml of medium, we could detect AHLs only at the inoculation spot (Fig 1E). Prolongation of the incubation time up to 48 hours changed the pattern of AHLs distribution in the medium. In the plates containing 7.5 ml of medium, where we observed the most efficient swarming of *Dickeya*, the radius of AHLs detection was reduced down to 2 cm from the inoculation spot (Fig 1F). In the plates with 10, 15, and 20 ml of the medium we detected AHLs at every tested spot (Fig 1G–1I). In the case of the plate containing 25 ml of medium AHLs were detected within 2 cm radius from the inoculation spot (Fig 1J).

Swarming motility is dependent on the metabolic state of bacteria and the availability of nutrients in the environment. We started with verifying the swarming of *Dickeya* on a medium

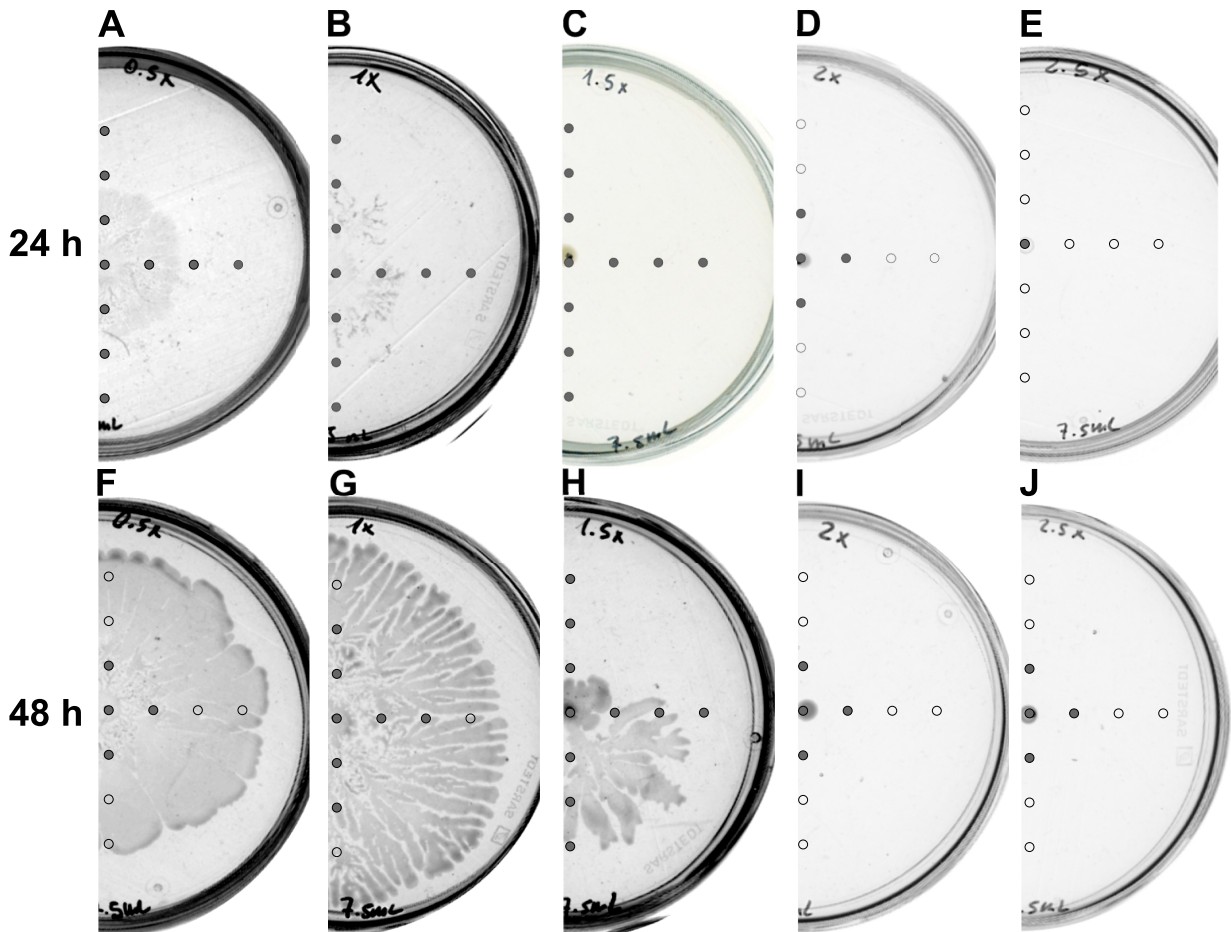

**Fig 2.** Swarming and AHLs production by *Dickeya solani* IFB102 on plates containing 0.5% of agar and the changing concentrations of B-medium: (A, F) 0.5x, (B, G) 1x, (C, H) 1.5x, (D, I) 2x, (E, J) 2.5x. Swarming assays were performed for 24 and 48 h, as indicated. Grey circles represent spots at which AHLs were detected. Open circles represent spots at which no AHLs were detected.

prepared with increasing content of its components while keeping agar concentration at 0.5%. We observed that after 24 h of incubation *Dickeya* swarmed on 0.5x and 1x concentrated B-medium (Fig 2A and 2B).

For higher concentrations of medium bacteria exhibited no swarming motility (Fig 2C–2E). Prolongation of incubation time up to 48 h enabled *Dickeya* to additionally start swarming on plates containing 1.5x concentrated medium (Fig 2C). The pattern of AHLs distribution in the medium changed along with the increase of medium concentration and incubation time. For 24 hours incubation we detected AHL molecules all over the plates containing 0.5x, 1x, and 1.5x medium (Fig 2A–2C). In the case of plates with 2x concentrated medium, we could detect AHLs within 1 cm radius from the inoculation spot (Fig 2D). Sampling of plates containing 2.5x concentrated medium allowed us to detect AHLs only at the inoculation spot (Fig 2E). The 48-hour incubation changed the distribution of AHLs in the tested plates. In the plates containing the least concentrated medium (0.5x), we detected AHL molecules within 1 cm radius from the inoculation spot (Fig 2F). The radius of AHLs detection increased up to 2 cm in the case of plates with 1x medium (Fig 2G) and up to at least 3 cm from the inoculation spot as observed for the plates containing 1.5x medium (Fig 2H). Further increase in medium

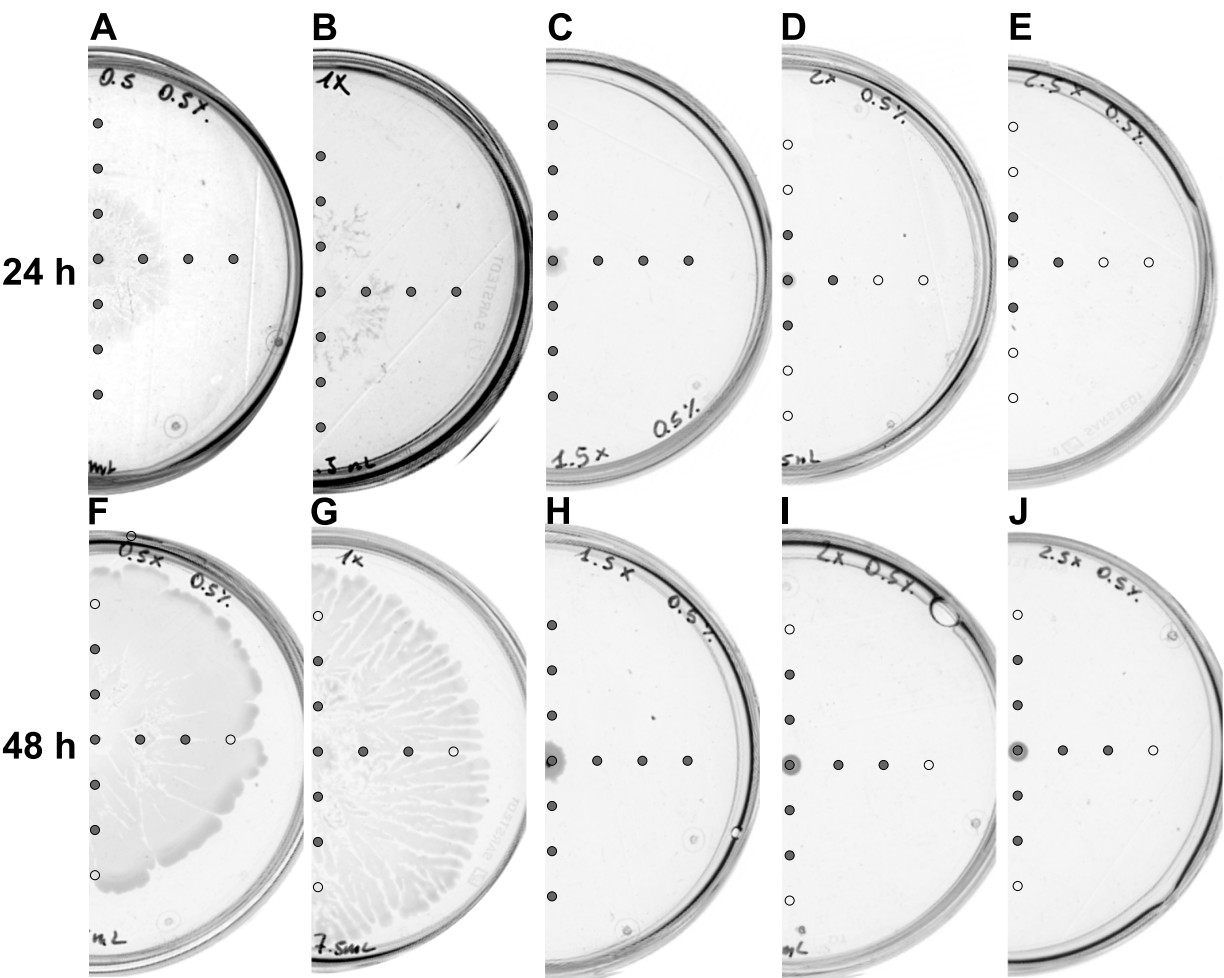

**Fig 3.** Swarming and AHLs production by *Dickeya solani* IFB102 on plates containing 0.5% of glucose, 0.5% of agar and the following concentrations of remaining components of B-medium: (A, F) 0.5x, (B, G) 1x, (C, H) 1.5x, (D, I) 2x, (E, J) 2.5x. Swarming assays were performed for 24 and 48 h, as indicated. Grey circles represent spots at which AHLs were detected. Open circles represent spots at which no AHLs were detected.

concentration (2x, 2.5x) resulted in decreasing the radius of AHLs detection down to 1 cm from the inoculation spot (Fig 2I and 2J).

The glucose content in the medium was shown to affect the motility of bacteria [23]. In our experiments with increasing concentration of medium components, one of them was this sugar. To assess whether observed changes in *Dickeya* swarming were due to the increased glucose concentration we performed a series of swarming assays on plates with a medium containing changing contents of its components while keeping glucose and agar concentrations at 0.5%. After both, 24 h and 48 h of incubation, we observed swarming of *Dickeya* only on plates containing 0.5x (Fig 3A and 3F) and 1x (Fig 3B and 3G) concentrated medium.

The pattern of AHLs distribution in the medium also changed along with the increasing concentration of medium components and incubation time. In the case of the 24-hour incubation, AHL molecules were detected in the entire plates containing 0.5x, 1x, and 1.5x concentrated medium with constant content of 0.5% glucose (Fig 3A–3C). It is worth noticing that on the plate containing 1.5x medium bacteria were not swarming. In the case of 2x and 2.5x concentrated medium with 0.5% glucose AHLs were detected within 1 cm radius from the

inoculation spot (Fig 3D and 3E). The 48-hour incubation enabled the detection of AHL molecules within 2cm radius from the inoculation spot in all plates apart from the plate containing 1.5x medium where AHLs detection radius increased up to 3 cm. Obtained results suggest that the swarming of *Dickeya* is affected not only by increasing glucose content but also other components of the B-medium.

In the final experiment, we wanted to analyze the influence of increasing glucose content on the swarming motility of *Dickeya*. Therefore, we performed a series of 24-hour swarming assays on 0.5x concentrated B-medium with glucose content increasing from 0% to 5%. In the results we could distinguish four different patterns of *Dickeya* swarming: (i) a small central colony without visible swarming (0% to 0.1% of glucose, Fig 4A–4D), (ii) a central colony with the increasing ring of swarming bacteria (0.25% to 0.4% of glucose, Fig 4E–4G), (iii) a small colony at the inoculation spot with extending irregular dendrites (0.5% to 3% of glucose, Fig 4H–4M), (iv) a large uniform central colony without dendrites (4% and 5% of glucose, Fig 4N and 4O). Obtained results suggest that some minimal concentration of glucose (at least 0.25%) was required to trigger *Dickeya* swarming motility. High glucose content (4% and 5%) was also not optimal for *Dickeya* to develop an efficient swarming. The distribution of AHL molecules in the medium depended on glucose content. For the lowest glucose contents (0%, 0.01% and 0.02%) the radius of AHLs detection was 2 cm from the inoculation spot (Fig 4A–4C).

For plates containing between 0.1% and 1% of glucose, AHLs were present within the radius of at least 3 cm from the inoculation spot (Fig 4D–4I). In the case of plates in which glucose content was between 1.5% and 3%, AHL molecules were detected in the area of medium covered with swarming bacteria (Fig 4J–4M). In the case of plates with the highest contents of glucose (4% and 5%), we were unable to detect any AHL molecules at all tested spots (Fig 4N and 4O).

## Discussion

Plant pathogens, for most of their lives, reside outside of their host. The efficient colonization of the plant required the development of mechanisms enabling sensing and active movement of pathogen cells towards favorable sites of infection. Upon adhesion to the plant tissues, bacteria can penetrate them through wounds and natural openings, thus establishing a successful infection. These mechanisms are chemotaxis and motility. Virulence of different plant pathogens, including *Ralstonia solanacearum*, *Pseudomonas phaseolicola*, *Pseudomonas syringae*, and *Dickeya dadanti*, have been demonstrated to be strictly dependent on the ability of bacteria to sense and actively move in the environment [24–26].

Genes involved in the motility regulation belong to the group exhibiting significant induction in the plant tissue [27]. Variations in the motility of *Dickeya solani* have also been proposed to contribute to its aggressiveness variability [28]. These facts clearly suggest the importance of motility in the virulence of *Dickeya* and their capability of efficiently infecting the plant host. It also needs to be emphasized that the motility of *D. solani* is dependent on AHLs-based quorum sensing [15], and therefore should be regarded as the social behavior of these bacteria.

Our observation that the swarming of *D. solani* on agar plates depends on the volume of used the medium (Fig 1) can be explained by two hypotheses. The first assumes that a small volume of medium in the plate results in a relatively fast depletion of available nutrients. In response to such conditions, bacteria could trigger swarming motility to enable the optimal dissemination of cells, as the local population was too large for a given niche [29]. An alternative hypothesis assumes that a smaller volume of medium in the plate enables faster accumulation of factors secreted by bacteria. Among these factors, AHL quorum sensing molecules

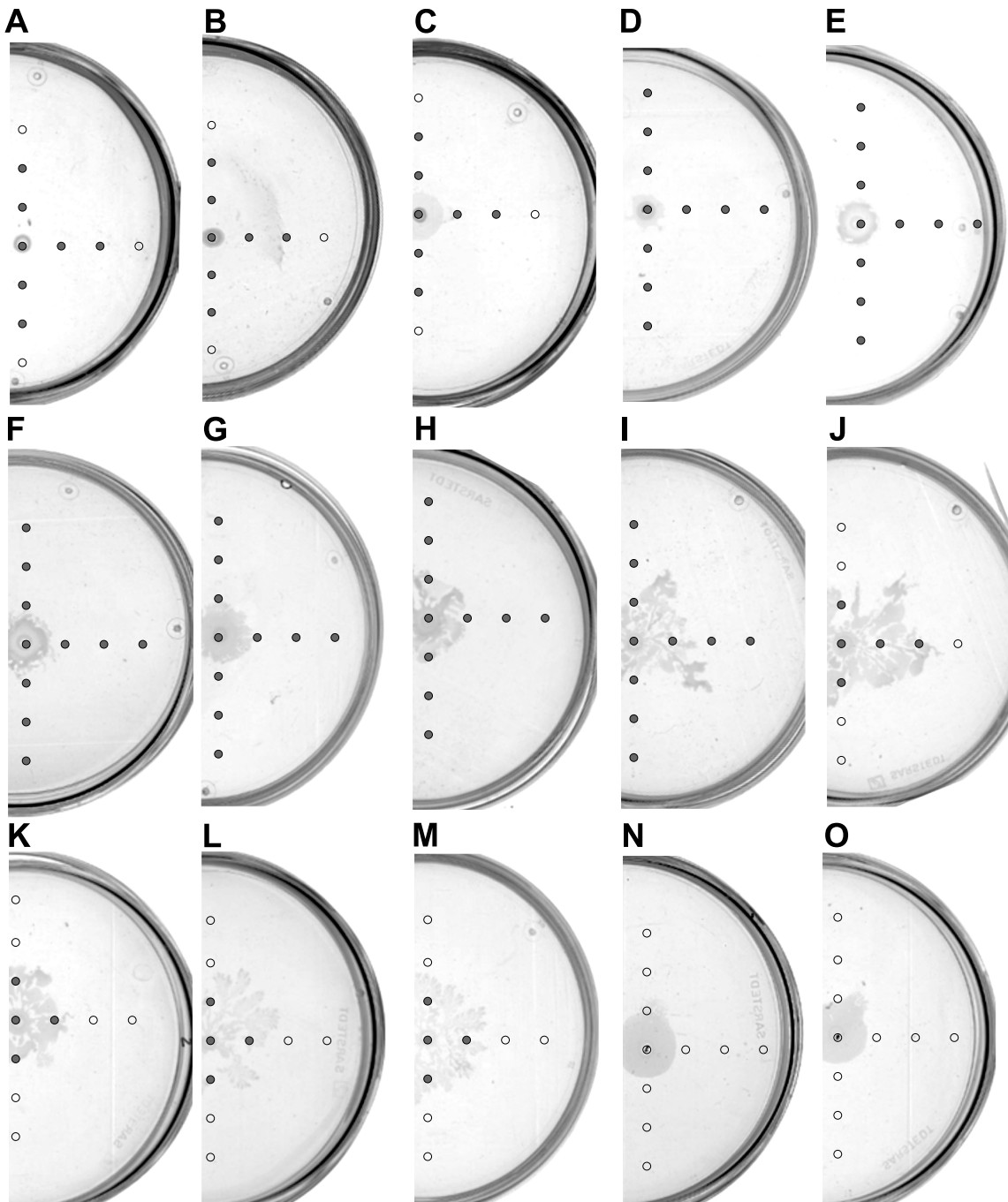

**Fig 4.** Swarming and AHLs production by *Dickeya solani* IFB102 on plates with B-medium containing 0.5% of agar and the following concentrations glucose: (A) 0%, (B) 0.01%, (C) 0.02%, (D) 0.1%, (E) 0.25%, (F) 0.3%, (G) 0.4%, (H) 0.5%, (I) 1%, (J) 1.5%, (K) 2%, (L) 2.5%, (M) 3%, (N) 4%, (O) 5%. Swarming assays were performed for 24 h. Grey circles represent spots at which AHLs were detected. Open circles represent spots at which no AHLs were detected.

might be responsible for triggering the swarming motility once the threshold concentration is reached. Our results support this hypothesis because we could observe the increase in the area of the plate with detectable AHLs as the volume of the medium decreased (Fig 1).

The influence of glucose on the motility of bacteria has been demonstrated by different research groups. In the early study, Armitage *et al.* showed that 1% glucose present in the medium delayed swarming of *Proteus mirabilis*, an enterobacterium associated with infections of the urinary tract in humans [30]. The study performed on several species of enterobacteria suggested that glucose did not affect swimming motility [31]. Moreover, swarming but not swimming motility of *Pseudomonas aeruginosa*, is dependent on changing concentrations of glucose in the medium [32]. It is important to emphasize that swimming and swarming motilities are different phenomena. While swimming is a non-coordinated movement of bacteria through the water channels in the agar, swarming is a social phenomenon of bacterial movement over the solid surface, requiring specific conditions and precise regulation [12]. Jahid et al. showed that increasing concentration of glucose inhibited swarming of *Aeromonas hydrophila*, a pathogen of fish and amphibians. Moreover, these authors extended their research and analyzed the influence of glucose on biofilm formation and proteases production. They proposed a conceptual model in which observed effects resulted from glucose-dependent modulation of quorum sensing [23]. Our results are concordant with the above observations since we showed that the swarming of *D. solani* changed along with glucose concentration in the medium (Fig 4). A similar effect was observed in the experiment with a concentrated medium (Fig 2) and could result from a higher content of glucose. Along with glucose-dependent changes in the swarming pattern, we observed differences in the distribution of AHL quorum sensing molecules in the medium. This suggests a tight association between glucose concentration in the medium, swarming, and quorum sensing in *D. solani*. So far no direct link between these has been demonstrated for these bacteria, nevertheless the study of Potrykus et al. indicated that quorum sensing systems in *D. solani* influenced the maceration of plant tissue, the production of plant cell wall-degrading enzymes, as well as swarming motility [15].

The major mechanism governing the utilization of different carbon sources in bacteria is carbon catabolite repression (CCR). This mechanism optimizes the uptake of the most efficiently metabolized sugars available at the moment in the environment, and glucose is the most preferred one [19]. The CCR was shown to regulate the virulence of different pathogenic bacteria. In *Pseudomonas aeruginosa* two quorum sensing systems: AHL-based (*las* and *rhl*) and non-AHL-based (*qps* and *iqs*), interact with one another to control the expression of genes according to the density of the bacterial population [33]. Quorum sensing in these bacteria was also shown to contribute to their virulence [34], moreover it was demonstrated to be coupled with the CCR via protein quality control (PQC) proteases Lon and ClpXP [35]. The AHL-based quorum sensing systems *las* and *rhl* in *P. aeruginosa* are involved in the production of rhamnolipids, biosurfactants required for efficient swarming of these bacteria [36]. These systems were also shown to be controlled by the CCR [35]. In the case of plant pathogens, the expression of genes involved in the virulence of *Dickeya dadanti* was demonstrated to be regulated by the CCR [20].

To conclude, the regulation of *D. solani* swarming and AHLs production by glucose seems to be via the carbon catabolite repression. Increasing concentration of glucose in the medium ensures availability of the most efficiently metabolizable sugar and thus turns the swarming motility into the unnecessary expense of energy. This cessation of motility is associated with decreased production of AHLs, and one can hypothesize that this low abundance of quorum sensing signaling molecules in the medium prevents the triggering of swarming. Results presented in our study broaden our knowledge about the physiology of the important plant pathogen *D. solani*. Moreover, they might be useful from a practical point of view. The outcomes of swarming assays performed with this bacterium apparently can be dependent not only on such

conditions as medium composition, temperature, and humidity but also can change with the volume of medium in the plate used to cultivate bacteria.

## Materials and methods

### Bacterial strains and media

*Dickeya solani* strain IFB102 [37] was used as a wild-type strain. Bacteria capability to produce and release acyl-HLs in the environment was assessed based on the reporter strain *Escherichia coli* [pSB401] (*Tc*[R]; *luxRI'*::*luxCDABE*) developed by Winson et al. [22]. Bacteria were cultured in Luria broth (LB) medium (tryptose 10 g/l, yeast extract 5 g/l, NaCl 10 g/l) supplemented with antibiotic when required. The temperature of growth was set at 28˚C for *D. solani* and at 37˚C in the case of *E. coli*. Swarming motility was performed on synthetic B-medium [38] which contains 15 mM $(NH_4)_2SO_4$, 8 mM $MgSO_4$, 27 mM KCl, 7 mM sodium citrate, 50 mM Tris/HCl (pH 7.5) supplemented on the day of inoculation with 0.6 mM $KH_2PO_4$ 2 mM $CaCl_2$, 1 µM $FeSO_4$, 10 µM $MnSO_4$, 4.5 mM glutamic acid, 0.78 mM tryptophan, 0.8 mM Lysine and 0.5% (w/v) glucose. All plates were prepared by supplementing the medium with 0.5% (w/v) of Bacto agar.

### Swarming motility

A single colony of *D. solani* IFB102 was inoculated in LB medium and incubated overnight with shaking at 28˚C. Two microliters of the overnight culture (OD600 $\approx$ 0.8) were inoculated in the center of a B-medium plate (0.5% of agar) and incubated for 24h or 48h at 28˚C (relative humidity 55% saturation). Plates were prepared 1 h before the inoculation and dried open for 30 min in a laminar flow chamber. B-medium plates contained different volume of medium (7.5 ml, 10 ml, 15 ml, 20 ml and 25 ml) and increasing concentration of medium (0.5x, 1x, 1.5x, 2x and 2.5x). Glucose influence on swarming motility was determined with plates containing 7.5 ml of 0.5x concentrated B-medium (0.5% of agar) with increasing concentration of glucose (0, 0.01, 0.02, 0.1, 0.25, 0.3, 0.4, 0.5, 1, 1.5, 2, 2.5, 3, 4 and 5% (w/v)).

### Detection of AHLs released in B-medium agar plates

In this study, we developed a fast-screening method for the detection of N-Acyl homoserine lactones (AHLs) in agar plates based on the method previously presented by Jafra et al. [39]. *Escherichia coli* [pSB401] was used as a biosensor for detection of AHLs due to its high level of bioluminescence mediated by the presence of N-3-(Oxohexanoyl)-L-homoserine lactones [22]. A single colony of *E. coli* [pSB401] was inoculated in 5 ml of LB supplemented with 20 µl/ml of tetracycline and incubated overnight at 37˚C. The day after, the culture was diluted to OD600 $\approx$ 0.1 and incubated for 5 hours. Swarming motility plates were examined to determine the presence of AHLs. Thirteen holes of $\sim$9 mm diameter were cut from each plate by using a flamed cork borer. The circular agar pieces were collected at 1 cm away from each other, starting from the inoculation point and proceeding in the four directions (above, below, left, right) up to 3 cm away from the point of inoculation. The circular agar samples were transferred directly to a sterile 96 wells plate and each well was inoculated with 150 µl of diluted suspension of the indicator strain (OD600 $\approx$ 0.2). Plates were incubated overnight at 37˚C, the growth temperature not permissive for *D. solani*. Emission of chemiluminescence was detected with the ChemiDoc XRS+ system (BIO-RAD).

## Author Contributions

**Conceptualization:** Adam Iwanicki, Michał Obuchowski.

**Data curation:** Roberta Gatta, Adam Iwanicki.

**Formal analysis:** Michał Obuchowski.

**Funding acquisition:** Michał Obuchowski.

**Investigation:** Roberta Gatta, Andrzej Wiese.

**Methodology:** Roberta Gatta, Andrzej Wiese.

**Project administration:** Michał Obuchowski.

**Supervision:** Michał Obuchowski.

**Validation:** Roberta Gatta, Adam Iwanicki.

**Visualization:** Adam Iwanicki.

**Writing – original draft:** Adam Iwanicki.

**Writing – review & editing:** Roberta Gatta, Adam Iwanicki, Michał Obuchowski.

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
