## [Decision Letter · Decision Letter 0]

3 Aug 2021

PONE-D-21-19894

Influence of glucose on swarming and quorum sensing of Dickeya solani

PLOS ONE

Dear Dr. Obuchowski,

Thank you for submitting your manuscript to PLOS ONE. After careful consideration, we feel that it has merit but does not fully meet PLOS ONE’s publication criteria as it currently stands. Therefore, we invite you to submit a revised version of the manuscript that addresses the points raised during the review process.

We look forward to receiving your revised manuscript.

Kind regards,

Manjunath Hubballi, Ph D

Academic Editor

PLOS ONE

2.Thank you for stating the following financial disclosure:

 “This work was supported by the National Science Center OPUS project no. 2018/29/B/NZ9/02339”

Please include this amended Role of Funder statement in your cover letter; we will change the online submission form on your behalf."

“This work was supported by the National Science Center OPUS project no. 2018/29/B/NZ9/02339”

 “This work was supported by the National Science Center OPUS project no. 2018/29/B/NZ9/02339”

Additional Editor Comments (if provided):

The changes suggested by reviewers can be made and resubmitted .

Reviewers' comments:

Reviewer's Responses to Questions

**Comments to the Author**

1. Is the manuscript technically sound, and do the data support the conclusions?

Reviewer #1: Partly

Reviewer #2: No

Reviewer #3: Yes

2. Has the statistical analysis been performed appropriately and rigorously? 

Reviewer #1: I Don't Know

Reviewer #2: No

Reviewer #3: Yes

3. Have the authors made all data underlying the findings in their manuscript fully available?

Reviewer #1: Yes

Reviewer #2: No

Reviewer #3: Yes

4. Is the manuscript presented in an intelligible fashion and written in standard English?

Reviewer #1: No

Reviewer #2: Yes

Reviewer #3: Yes

5. Review Comments to the Author

Reviewer #1: manuscript need to revision.

Please clearly explain the aims of the results and study…

there were some grammar or writing mistakes..

you can add more figures..

33 signaling molecules are different depending on the volume of the medium in the plate.

In this study, we show that the swarming motility of Dickeya

41 solani and production of the quorum-sensing molecules are coupled with the central ….

44 Introduction

64

65 towards their source, is most probably ….

92 for 24 and 48 has indicated.

The 48-hour incubation enabled the detection of AHL

204 bacteria can penetrate them through wounds and natural openings,

219 response to such conditions, bacteria could trigger swarming motility to enable optimally

232 swarming but not swimming motility of Pseudomonas aeruginosa, is

233dependent on changing concentrations of glucose in the medium [32]. um

271 sensing signaling molecules in the medium prevent the triggering of swarming.

Materials and Methods

278 Bacterial strains and media

279 Dickeya solani strain IFB102 (21) was used as a wild-type strain.

Reviewer #2: The novelty of the work is questionable. The extent of experimentation is not what it should be to reach a valid conclusion.

The dataset is very limited, which makes it not suitable for further consideration for publication.

Reviewer #3: Study on infection process of pathogen Dickeya solani in potato plants is very interesting. Clear understanding of host-pathogen interaction process was taken in this study. The author proved that how pathogen are attracted to the infection site and justified through the simple technique. It is worth to publish

6. PLOS authors have the option to publish the peer review history of their article (what does this mean?). If published, this will include your full peer review and any attached files.

Reviewer #1: No

Reviewer #2: No

Reviewer #3: **Yes: **Dr. A. Ramanathan, Professor (Plant Pathology)

---

## [Author Response · Author response to Decision Letter 0]

10 Aug 2021

Specific answers to reviewers:

Reviewer #1:

The aims of the study have been clearly explained in the abstract section (lines 28-31).

The manuscript has been corrected according to Reviewer’s suggestions (lines: 33, 42, 66, 102, 134, 162, 170, 205, 220-221,272, 280).

Reviewer #2: 

To our knowledge, this is the first study performed on Dickeya solani in which swarming motility of this bacterium was shown to depend on the volume on the medium in the cultivation plate and glucose content. While the influence of the analyzed sugar on the swarming motility was shown by other authors in context of other bacterial species there are no studies documenting such phenomenon in the case of D. solani.

Reviewer #3:

We thank very much for kind appreciation of our work.

---

## [Editor Report · Decision Letter 1]

13 Jan 2022

Influence of glucose on swarming and quorum sensing of Dickeya solani

PONE-D-21-19894R1

Dear Dr. Michał Obuchowski,

We’re pleased to inform you that your manuscript has been judged scientifically suitable for publication and will be formally accepted for publication once it meets all outstanding technical requirements.

Kind regards,

Abdelwahab Omri, Pharm B, Ph.D

Academic Editor

PLOS ONE
---

## [Editor Report · Acceptance letter]

11 Feb 2022

PONE-D-21-19894R1 

Influence of glucose on swarming and quorum sensing of *Dickeya solani*

Dear Dr. Obuchowski:

I'm pleased to inform you that your manuscript has been deemed suitable for publication in PLOS ONE. Congratulations! Your manuscript is now with our production department. 

Kind regards, 

on behalf of

Dr. Abdelwahab Omri 

Academic Editor

PLOS ONE